# Study on the Effect of Pre-Refinement and Heat Treatment on the Microstructure and Properties of Hypoeutectic Al-Si-Mg Alloy

**DOI:** 10.3390/ma15176056

**Published:** 2022-09-01

**Authors:** Ling Lin, Lian Zhou, Yu Xie, Weimin Bai, Faguo Li, Ying Xie, Mingxin Lu, Jue Wang

**Affiliations:** 1School of Materials Science and Engineering, Xiangtan University, Xiangtan 411105, China; 2Central Research Institute, Baoshan Iron & Steel Co., Ltd., Shanghai 201900, China

**Keywords:** Al-Si-Mg alloy, pre-refinement, grain refinement, microstructure, mechanical properties

## Abstract

Hypoeutectic Al-Si-Mg alloys with a silicon content of around 10 wt % are widely used in the aerospace and automotive fields due to their excellent casting properties. However, the occurrence of “silicon poisoning” weakens the refinement effect of a conventional refiner system such as Al-5Ti-1B. In this paper, we proposed the “pre-refinement” method to avoid the “Si poisoning” to recover the refinement effect of Al-5Ti-1B. The core concept was to adjust the order of adding the Si element to form the TiAl_3_ before forming the Ti-Si intermetallic compound. To prove the effectiveness of the “pre-refinement” method, three alloys of “pre-refinement”, “post-refinement”, and “non-refinement” of an Al-10Si-0.48Mg alloy were prepared and characterized in as-cast and heat-treatment states. The results showed that the average grain diameter of the “pre-refinement” alloy was 60.19% smaller than that of the “post-refinement” one and 81.34% smaller than that of the “non-refinement” one, which demonstrated that the proposed method could effectively avoid the “silicon poisoning” effect. Based on a refined grain size, the “pre-refinement” Al-10Si-0.48Mg alloy showed the best optimization effect in mechanical properties after a solid-solution and subsequent aging heat treatments. The best mechanical properties were found in the “pre-refinement” alloy with 2 h of solid solution treatment and 10 h of aging treatment: a hardness of 92 HV, a tensile strength of 212 MPa, and an elongation of 20%.

## 1. Introduction

Al-Si-Mg casting alloys have been widely used in the aerospace, automotive, and 3C industries due to their excellent casting properties and outstanding strength-to-weight ratio. With the continuous development of the new-energy automotive industry, the use of Al-Si-Mg casting alloys will be further increased. The excellent performance of Al-Si-Mg casting alloys depend largely on the microstructure, which is characterized by the grain size of the matrix phase α-Al and the shape and distribution of the eutectic phase comprising silicon and Mg_2_Si [1,2]. Grain refinement can produce more equiaxed crystal structures and reduce casting defects, and is an effective method to simultaneously improve the strength and toughness of casting alloys [3]. 

As early as 1930, Rosenhain et al. [4] found that adding the Ti element during the smelting process of Al alloys can refine the grain size, and related research on grain refinement has been conducted since then. Jones and Pearson [5] believe that the best grain-refining capacity needs a period of time to emerge; however, if the standing time during the casting is set too long, the grain size become coarser again. The decline in the refinement effect is attributed to the aggregation and precipitation of TiB_2_ and TiC particles or the loss of their activity due to the change in the surface structure [6]. Grain refinement by adding refiners such as Al-Ti-B and Al-Ti-C is the conventional method to obtain fine grains for Al-Cu and Al-Mg alloys. Some alloying elements (such as Fe, Si, Mg) can improve the refining effect of grain refiners. However, when the alloying element content exceeds a certain value, the refining effect of the refiner decreases with the increase in the alloying element content [7,8]. However, for Al-Si alloys, it has been found that the refinement effect of the commonly used refiners is weakened when the content of Si element exceeds 3.5 wt %, which is known as the “Si poisoning” effect [9,10]. In the past, it was widely believed that the alloying elements with a toxic effect on the intermediate alloy would concentrate around TiB_2_ or TiC particles and react with the Ti element, changing the surface physical and chemical properties of TiB_2_ or TiC particles, reducing the surface activity of TiB_2_ or TiC particles, and making the interfacial compatibility with melted aluminum worse. Schumacher and McKay found that the formation of TiSi_2_ on basal faces of TiB_2_ reduced the nucleation area and number of active sites for Al [11]. The reduction inα-Al growth restriction and the formation of silicide phases before solidification ofα-Al was due to a strong exothermic interaction between titanium and silicon, which is the main mechanism of Si poisoning [12]. In fact, the small degree of grain refinement by additions of eutectic-forming elements (Cu, Mg and Si) is mainly attributed to their segregating power; however, they cannot form nucleant particles [13].The nucleation work of α-Al on its surface increases and the grain refinement effect decreases [14,15]. It should be noted that poisoning occurs whether or not it is grain-refined with Al-Ti-B master alloys [16]. Therefore, we must find a way to circumvent the influence of Si on the nucleation and growth of α-Al.

Upon the addition of Al-5Ti-B refiner, the bulk TiAl_3_ dissolves into the liquid, leaving TiB_2_ as the nucleation particle for α-Al [17], then the Si segregates on the surface of TiB_2_ particles and dissolves into a TiAl_3_ two-dimensional compound (TiAl_3_ 2DC). The TiAl_3_ 2DC plays a key role in nucleation, and then the strong interaction between the Si atoms and Ti atoms can disturb the crystal structure of TiAl_3_ 2DC to a certain extent, which as a result weakens the nucleation efficiency, and the so-called “Si-poisoning” occurs. Alternatively, an Al-Nb-Ti-B intermediate alloy rich in (Nb,Ti)B_2_ particles with a “sandwich” structure is expected to be an effective refiner for high-silicon aluminum alloys [18]. Unfortunately, the addition of Nb elements also has the disadvantages of increasing the cost and the difficulty of accurate preparation. 

The key of TiB_2_ as the nucleation core of α-Al is to promise the function of the Ti-terminated (0001) surface as the nucleation template [19,20]. The “Si poisoning” lies in that the preferential combining of Si atoms and Ti atoms, which forms Ti-Si intermetallic compounds, destroys the Ti-terminated (0001) surface. Naturally, the “Si poisoning” can probably be avoided by adjusting the order of the addition of alloying elements into the Al alloy to ensure the Al atoms combine with Ti atoms before Si atoms. Studies have found [21,22] that the liquid Al atoms can form an orderly layered structure in front of the solid Ti atoms around the melting point of pure aluminum, which can benefit the heterogeneous nucleation. In this paper, a concept of “pre-refinement” is proposed to recover the refinement effect of the conventional Al-5Ti-B refiner for the hypoeutectic Al-Si-Mg alloy. The core concept was to adjust the order of adding the Si element to form the TiAl_3_ before forming the Ti-Si intermetallic compound. The mechanical properties of the Al alloy were further improved through the subsequent heat-treatment process upon the grain-refining strengthening. 

## 2. The Concept of the “Pre-Refinement” Design

The basic principle of “pre-refinement” was to ensure that the liquid Al atoms near the melting point of aluminum could form an orderly layered structure in front of solid Ti atoms. To satisfy this principle, the Al-5Ti-B refiner was firstly added to the Al-Mg alloy at a temperature close to the melting point of the alloy to ensure that Al atoms were orderly arranged on the Ti-terminated (0001) surface of the TiB_2_ particles to form the TiAl_3_ precursor. Si was then added, and as schemed, the Si atoms would not combine with Ti atoms to form a Ti-Si intermetallic compound due to the isolation of the Al atoms. During the subsequent cooling process, TiAl_3_ formed around TiB_2_ particles and α-Al grew up continuously on the TiAl_3_ substrate, achieving heterogeneous nucleation to refine the grains. The schematic of the “pre-refinement” is shown in Figure 1.

Based on ZL 104 (ZAlSi9Mg, Chinese grade of aluminum alloy, 8.0–10.5 wt % Si, 0.2–0.5 wt % Mn, 0.17–0.35 wt % Mg), we expected to develop a new Al-10Si-0.48Mg alloy for automotive die casting integration. However, the Si content was more than 3.5 wt %, so the conventional method of adding an Al-Ti-B refiner could not produce the refining effect. Therefore, we attempted the “pre-refinement” method using the alloy Al-10Si-0.48Mg.

We used the results of thermodynamic calculation to help design the key parameters during the pre-refinement and the following heat-treatment process. The equilibrium solidification phase diagram of the Al-10Si-0.48Mg was calculated using the built-in database in the trial version of Thermo_Calc (Figure 2a; the database was restored according to reference [23]). The precipitation temperatures of the α-Al, Si, and Mg_2_Si phases were 593.63, 574.57, and 506.94 °C, respectively. The liquid phase vanishing temperature was 565.13 °C. The percentage of each equilibrium phase at room temperature was as follows: 89.829% α-Al, 9.373% Si, and 0.802% Mg_2_Si. The solidification paths of the Al-10Si-0.48Mg using the equilibrium and Scheil schemes (non-equilibrium solidification) were calculated using the trial version of Thermo_Calc (Figure 2b). The Mg_2_Si precipitated at 559 °C of the non-equilibrium solidification. The final solidification temperature was 6.5 °C lower than that of the equilibrium solidification. 

## 3. Experimental Materials and Methods

The Al-10Si-0.48Mg alloy was casted in the experiment by using materials including high-purity Al strips (99.999 wt %), high-purity Mg particles (99.99 wt %), high-purity Al foil (99.999 wt %), high-purity Si blocks (99.999 wt %), an Al-5Ti-1B refiner, and a covering agent (50 wt % NaCl + 35 wt % KCl + 15 wt % NaF). The final smelted aluminum ingot size was Φ40 × 50 mm. In the study, the macro and micro structure images were sampled in the middle of the ingot. The average grain size value was the average value of a total of 6 grain size values taken at the axial directions of 10, 25, and 40 mm and the radial directions of 5, 10, and 15 mm. To prove the effectiveness of “pre-refinement” design, the “pre-refinement”, “post-refinement”, and “non-refinement” Al-10Si-0.48Mg alloys were prepared. The preparation paths of the three types of alloys are shown in Figure 3. In the following, we will use the example of the pre-refinement preparation process to provide more details on the preparation process. During the preparation, the Al strips were put into the alumina ceramic crucible and melted at 700 °C, 33 °C higher than the melting point, in a well furnace (Xiangtan Samsung Instrument Co., Ltd., Xiangtan, China). Then, the furnace temperature was decreased to 675 °C and was held to add other materials. To prevent accidents such as fire caused by direct contact between Mg particles and the high-temperature liquid Al, the Mg particles were wrapped with aluminum foil and placed in the liquid. Then, the Al-5Ti-1B refiner was added into the Al-Mg liquid; it was necessary to gently stir with a graphite rod to ensure the refiner was evenly distributed in the alloy liquid to increase the refinement effect. After about 10 min, silicon blocks were pressed into the alloy liquid with a graphite rod so that the silicon blocks could be quickly and evenly dissolved in the alloy liquid. Finally, the liquid was casted into a graphite crucible that was quenched in air to room temperature. 

A solution heat treatment at 500–530 °C for 2–8 h after water quenching and artificial aging at 170–210 °C for 2–20 h are the commonly used heat treatments to improve the mechanical properties of alloys [24,25]. Likewise, the T6 heat-treatment process of a solid solution treatment followed by an aging treatment was carried out for the three types of as-cast alloys.

According to the calculation results shown in Figure 2, the temperatures of the solid solution treatment and aging treatment were 500 °C and 175 °C, respectively. The time of the solid solution treatment was 2 h or 8 h, and the time of the aging treatment was 10 h or 20 h. Thus, four groups of heat-treatment processes were used: 2 h solid solution treatment + 10 h aging treatment (denoted as 2 + 10), 2 h solid solution treatment + 20 h aging treatment (denoted as 2 + 20), 8 h solid solution treatment + 10 h aging treatment (denoted as 8 + 10), and 8 h solid solution treatment + 20 h aging treatment (denoted as 8 + 20). The roadmap for heat treatment is show in Figure 4.

The microstructure and composition distribution were characterized using OM (ZEISS, Zeiss, Jena, Germany), SEM with EDS (ZEISS EVO MA10, Zeiss, Jena, Germany), and XRD (Ultima IV, Rigaku Co., Tokyo, Japan). The microstructure hardness and tensile tests were carried out using a Vickers hardness tester (SHYCHVT-30, Laizhou Huayin hardness meter factory, Laizhou, China) and an electronic multifunctional stretching machine (WDW-100C, Jinan Fangyuan Instrument Co., Ltd., Jinan, China), respectively. The dimensions of the tensile sample are shown in Figure 5 (in mm). 

## 4. Results and Discussion

### 4.1. Microstructures of the As-Cast and Heat-Treated Alloys

Figure 6a shows the SEM image of the as-cast “pre-refinement” hypoeutectic Al-10Si-0.48Mg alloy. EDS data showed that Point A, Point B, and Point C were α-Al with the Al:Mg:Si equal to 90.5:9.5:0, eutectic phase with Al:Mg:Si equal to 39.86:18.63:41.51, and Si + (Al + Si) phase with Al:Mg:Si equal to 20.59:0:79.41. The phase constituents of the “pre-refinement” hypoeutectic Al-10Si-0.48Mg alloy after 8 h solution treatment + 20 h aging treatment remained α-Al, Mg_2_Si, and Si phases, as shown in Figure 6b. 

The metallographic structure of the “post-refinement” hypoeutectic Al-10Si-0.48Mg alloy is shown in Figure 6c, in which the phase in the gray region is the α-Al matrix, the black particle phase is the Mg_2_Si intermetallic compound, and the needle block phase in the Al matrix is Si. It can be seen that the corner of the needle block was rounded. The α-Al phase was in the shape of large clusters with coarser grains compared with the pre-refinement one, indicating that the refinement effect of the Al-5Ti-1B refiner did not work well in the “post-refinement” preparation process. Due to the early addition of the Si element in the “post-refinement” method, the Ti and Si combined first, so the “silicon poisoning” phenomena was thus obvious. 

The metallographic structure of the “non-refinement” hypoeutectic Al-10Si-0.48Mg alloy is shown in Figure 6d, in which the grey region is the α-Al matrix, the black particle phase in the shape of Chinese character is the Mg_2_Si intermetallic compound, and the black line phase is the Al-Si eutectic. As can be seen in Figure 6d, the matrix had much coarser dendrites than the above two types of alloys. The Chinese-character Mg_2_Si phase had obvious angles appearing at the periphery, which can easily cause stress concentration in the preparation process and in service, and can damage the matrix and cause a large reduction in the mechanical properties of the alloy.

The macrostructures of the Al-10Si-0.48Mg alloy after different heat treatments are shown in Figure 7. The grain size distribution was uniform and the grain size of the pre-refinement sample was significantly smaller than that of the other two treatments. The grains of the pre-refinement samples showed fine equiaxed grains, the grains of the post-refinement samples showed coarse equiaxed grains, and the grains of the non-refinement samples showed coarse columnar grains.

The grain size of an Al alloy can be approximately measured as the dendrite size of α-Al, which has polyhedral equiaxed grains. Under a polarizing microscope, different orientation grains have different colors. Because the grain color distribution has a certain randomness, it indicates that the grain orientation obtained by the casting process is randomly arranged. Figure 8 shows the polarized microstructure and the average size of α-Al dendrites after different preparation methods and heat treatments. As shown in Figure 8, the average α-Al dendrite sizes of the “pre-refinement”, “post-refinement”, and “non-refinement” Al-10Si-0.48Mg alloys were 284.91 μm, 715.61 μm, and 1526.82 μm, respectively. The average grain size of samples after multiple casting can reflect the refining ability of the three kinds of smelting processes. It can be seen the “pre-refinement” process had an obvious refinement effect on the Al-10Si-0.48Mg alloy; the grain size was only 18.66% of that of “non-refinement”. Although the grain size of “post-refinement” was 46.87% of that of “non-refinement”, the grain-refining capacity of the Al-5Ti-1B was obviously weakened by the “silicon poisoning”. 

Compared with the effect of the casting preparation on the grain size, the solution and aging treatments had little effect on the grain size. However, the morphology was affected by the heat treatment. As shown in Figure 8, the morphology of α-Al changed significantly in the “pre-refinement” Al-10Si-0.48Mg alloy during the heat treatments. As the solid solution treatment proceeded, the size of eutectic silicon decreased and the morphology changed from needle to spherical. With further solid solution treatment from 2 h to 8 h, the size of eutectic silicon increased. During the solid solution treatment, the eutectic silicon underwent two processes with time, including fragmentation or dissolution and the spherization of separated branches. If the dissolution time was too long, obvious pores would occur, leading to excessive burning. The coarsening of the microstructural composition and possible pores would have a negative impact on the mechanical properties. The coarsening of spheroidal eutectic silicon α-Al could be observed in either the “pre-refinement” or the “post-refinement” alloys, while only the process of breaking of eutectic silicon could be observed in the “non-refinement” alloys during the treatment. 

### 4.2. Mechanical Properties of the As-Cast and Heat-Treated Alloys

Al-Si-Mg cast alloys are usually heat treated to obtain the best combination of strength and ductility. Solution treatment and the following aging treatment are the commonly used heat treatment tools, and precipitation hardening is the main mechanism during heat treatment of Al-Si-Mg alloys. For the Al-Mg_2_Si quasi-binary system, the Mg_2_Si phase precipitates as follows: SSS → GP zone → β″ → β′ → β (Mg_2_Si). These particles are invisible under the optical microscope, but this change can be indirectly observed with the change in alloy hardness and tensile strength due to the precipitates [26]. 

In this paper, the hardness of “pre-refinement”, “post-refinement”, and “non-refinement” as-cast and as-heated alloys of Al-10Si-0.48Mg alloy were tested. In the hardness test, six points on the surface of the sample were selected uniformly and randomly, and the hardness of the material was obtained by obtaining the average value of the six points. The measured hardness curve is shown in Figure 9. As can be seen in Figure 9, the effect of heat treatments was the most obvious in the “pre-refinement” alloy and the least in the “non-refinement” alloy. The highest hardness was achieved by the “pre-refinement” alloy with the 2 solid solution + 20 aging treatment. 

Some studies showed that increasing the aging temperature led to the higher solid solubility of the Si element in α solid solution with almost no change in the Mg element. Thus, the Si content at higher aging temperature exceeds the content needed to form the reinforcing phase Mg_2_Si, and the amount of precipitated Mg_2_Si phase is dependent on the Mg content but independent of aging temperature [27].

The hardness decreased after the solid solution treatment because the dissolving of the secondary phases occurred during the casting process and then gradually increased during the aging process due to the precipitation of the secondary phase. As can be seen in Figure 9, the hardness of “pre-refinement” reached a peak value of about 92 HV with 2 h solid solution + 20 h aging. This was called the peak aged (PA) state and was related to the formation of a metastable intermetallic. With the aging time continually increasing, the hardness decreased, which corresponded to an over-aged state that coarsened the precipitates, lost compatibility with the matrix, and gradually reduced the hardness value [28]. Likewise, the hardness had a similar change with the varied heat-treatment condition for the “post-refinement” alloy and the “non-refinement”, as shown by the curve in red and blue lines in Figure 9. However, it can be seen that the aging effect was small for the non-refinement alloys and the heat-treatment hardness was less than the as-cast hardness, indicating that few precipitates were produced during the aging. 

Figure 10 shows the tensile curves of the three cast alloys under different heat treatment processes of “pre-refinement”, “post-refinement”, and “non-refinement”. As can be seen in Figure 10a, the tensile strength of the hypoeutectic Al-10Si-0.48Mg alloy samples of each kind of alloy increased significantly after heat treatment and then gradually decreased the extension of aging time, which was in accord with the change in hardness and with the normal hardness–aging time relation. The elongation of the original hypoeutectic Al-10Si-0.48Mg alloy was not high for each kind of as-cast alloy due to the large secondary phases in the matrix, as shown in Figure 8. During the heat treatment, the large secondary phase dissolved into the matrix and then precipitated in a fine shape during aging, which benefited the elongation of the materials. It can be seen in Figure 10b that the elongation of the hypoeutectic Al-10Si-0.48Mg was increased by up to about 30%. The tensile strength of the alloy prepared using the “pre-refinement” method could reach about 212 MPa with 2 h solid solution + 10 h aging, which was a combination of a fine grain size and fine precipitates after the heat treatment. 

Compared with the effect of heat treatment on the increase in mechanical properties for the as-cast alloy, it can be seen the same heat treatment had a greater effect on the mechanical properties of the alloys with a finer grain. Therefore, obtaining a desirable as-cast microstructure is the first step in obtaining more excellent mechanical properties of alloys whether followed by the heat treatment or not. For example, the “pre-refinement” concept used to refine the grain of as-cast alloy in this paper promises excellent comprehensive mechanical properties for Al-Mg-Si alloys after heat treatment. 

## 5. Conclusions

In the paper, we proposed a concept of “pre-refinement” to avoid the “silicon poisoning” problem to refine hypoeutectic Al-Si alloys using a conventional refiner system such as Al-Ti-B. The core concept was to adjust the order of adding the Si element to form the TiAl_3_ before forming the Ti-Si intermetallic compound. The feasibility of this method was demonstrated by the contrast experiments on the “pre-refinement”, “post-refinement”, and “non-refinement” Al-10Si-0.48Mg alloys. The grain size of the as-cast “pre-refinement” alloy was much smaller than either the “post-refinement” one or the “non-refinement” one, demonstrating that the “pre-refinement” method could effectively avoid the “silicon poisoning” effect and recover the refinement effect of the conventional refiner system. Based on a refined grain, the “pre-refinement” Al-10Si-0.48Mg alloy showed the best optimization effect in mechanical properties upon a solid solution and subsequent aging heat treatment. The tensile strength of the alloy prepared using the “pre-refinement” method could reach about 212 MPa with 2 h solid solution + 10 h aging with a good elongation of 20% compared with the “non-refinement” one with a tensile strength of 112 MPa and an elongation of about 30%. 

## Figures and Tables

**Figure 1 materials-15-06056-f001:**
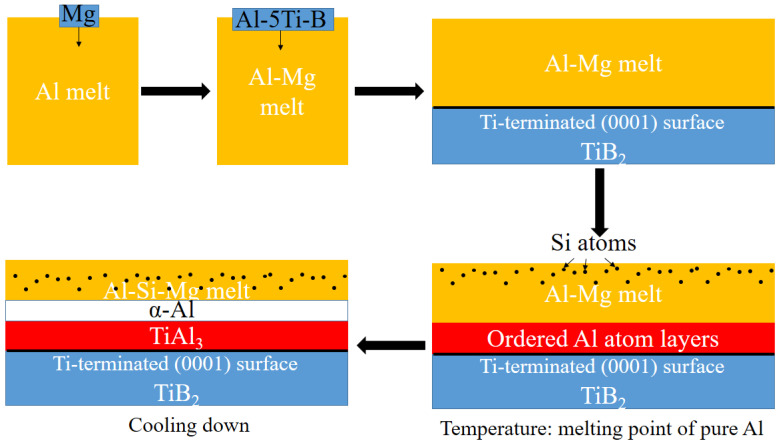
Schematic diagram of “pre-refinement” design concept.

**Figure 2 materials-15-06056-f002:**
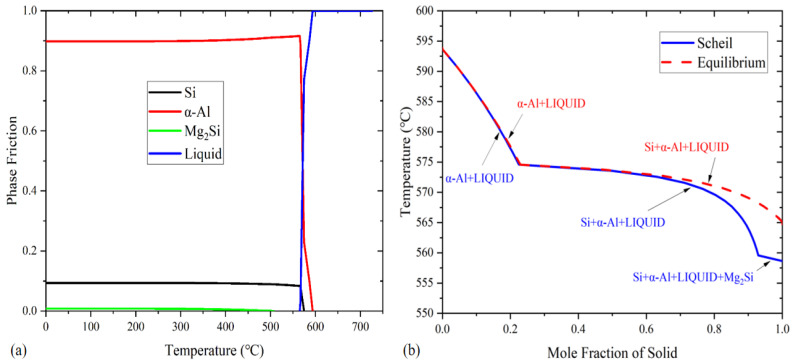
(**a**) Equilibrium solidification phase diagram for Al-10Si-0.48Mg; (**b**) calculated solidification paths for Al-10Si-0.48Mg according to the equilibrium and Scheil schemes.

**Figure 3 materials-15-06056-f003:**
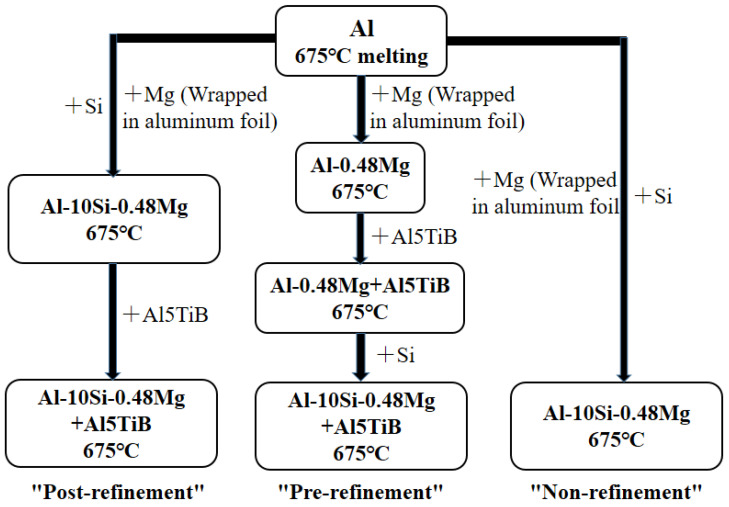
Schematic diagram of preparation paths of the three types of alloys: “post-refinement”, “pre-refinement”, and “non-refinement”.

**Figure 4 materials-15-06056-f004:**
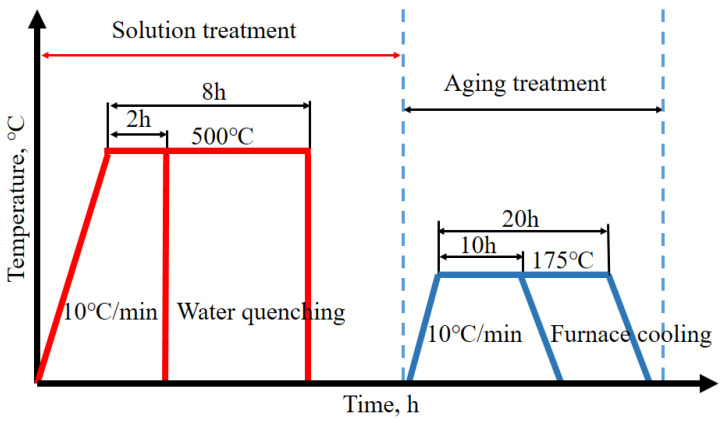
Roadmap for heat treatment.

**Figure 5 materials-15-06056-f005:**
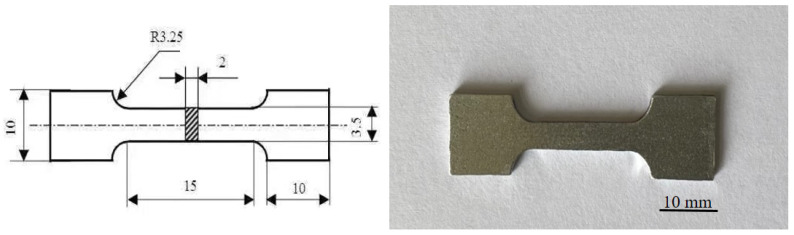
Dimensional drawing of tensile specimen and real specimen.

**Figure 6 materials-15-06056-f006:**
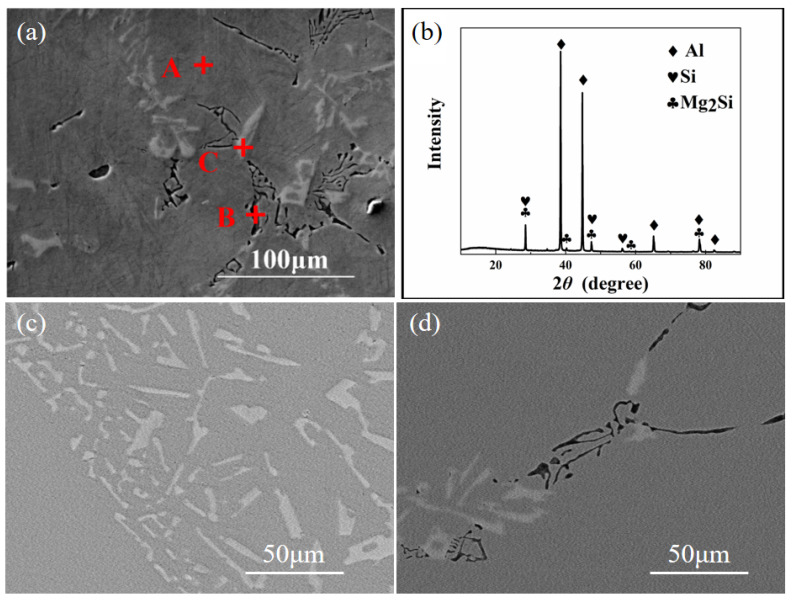
(**a**) SEM morphology of “pre-refinement” as-cast sample (A+, B+ and C+ are the positions of the spectral point scanning respectively); (**b**) XRD diffraction patterns of “pre-refinement” sample (8 + 20); (**c**) SEM morphology of “post-refinement” as-cast sample; (**d**) SEM morphology of “non-refinement” as-cast sample.

**Figure 7 materials-15-06056-f007:**
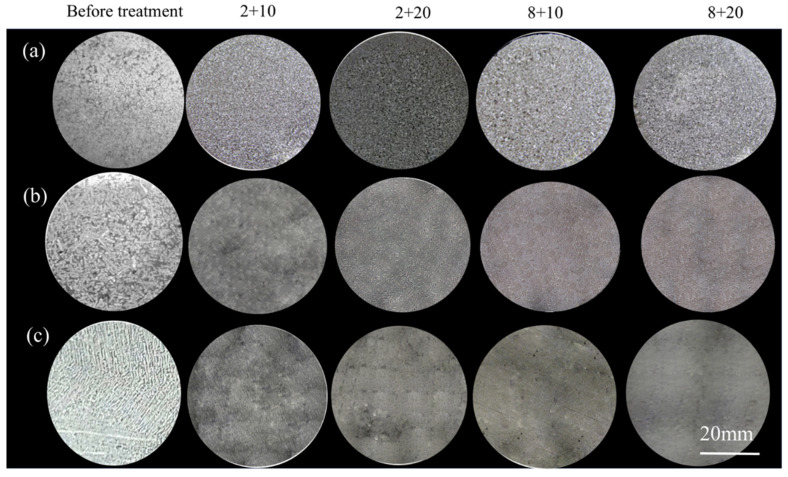
Macrostructures of Al-10Si-0.48Mg alloys after different heat treatments: (**a**) pre-refinement; (**b**) post-refinement; (**c**) non-refinement.

**Figure 8 materials-15-06056-f008:**
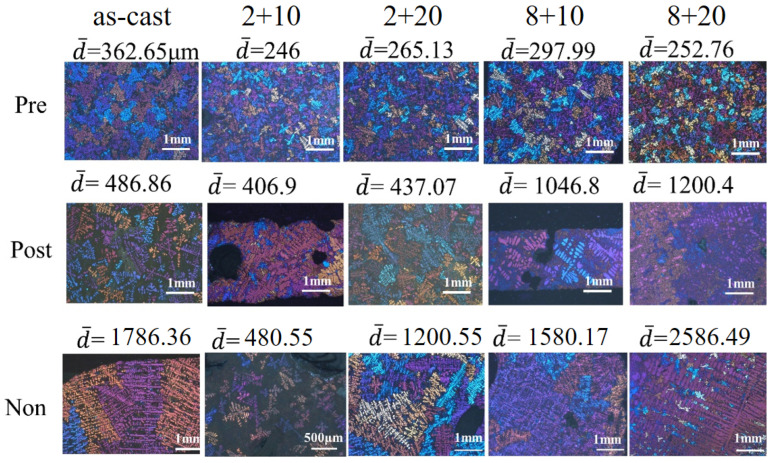
Dendrite structure and average grain diameter (μm) of the “pre-refinement”, “post-refinement”, and “non-refinement” as-cast and various heat-treated samples.

**Figure 9 materials-15-06056-f009:**
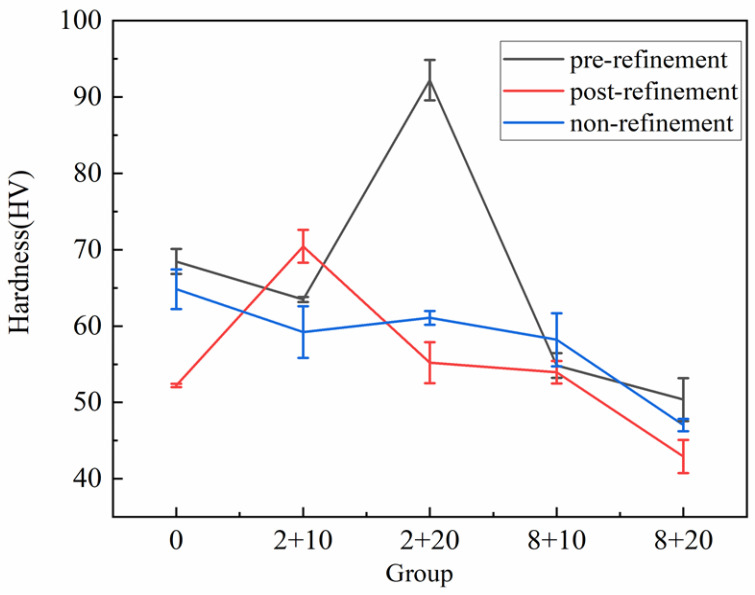
Hardness curves of “pre-refinement”, “post-refinement”, and “non-refinement” hypoeutectic Al-10Si-0.48Mg alloys with different heat-treatment methods.

**Figure 10 materials-15-06056-f010:**
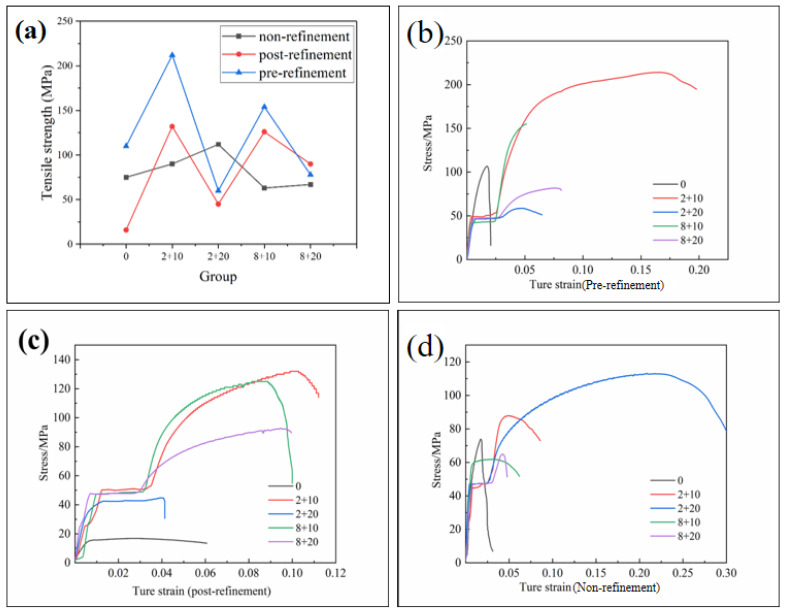
(**a**) Tensile strength of three casting alloys after “pre-refinement”, “post-refinement”, and “non-refinement” treatment; (**b**) tensile curves of “pre-refinement” casting alloys after different heat treatments; (**c**) tensile curves of “post-refinement” casting alloys after different heat treatments; (**d**) tensile curves of “non-refinement” casting alloys after different heat treatments.

## Data Availability

Not applicable.

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
