# Peer review of "Study on the Effect of Pre-Refinement and Heat Treatment on the Microstructure and Properties of Hypoeutectic Al-Si-Mg Alloy"

_materials, 2022, doi:10.3390/ma15176056_

Round 1

Reviewer 1 Report

This paper reports the characterization results of pre-refinement and heat treatment of Al-Si-Mg alloy. This topic is interesting and worthy to be investigated. However, this paper reports just simple characterization results without any scientific discussion based on thermodynamics or diffusion. This paper reports the phase equilibria, which must be discussed with phase diagram study.

However, no suitable phase diagram was suggested. In my opinion, a characterization of the Al-Si-Mg alloy after refinement and heat treatment should be performed to better highlight the grain evolution or in alternative EBSD maps. In this case, it is possible to have information’s concerning the grain shape and size. This will contribute to obtaining strong conclusions and highlights of work.

In addition, there are many scratches in the figure 5 (c, d), which demonstrates poor preparation of metallographic samples. It is very difficult to verify the validity of the analysis related to Fig. 6 because of the low quality and size of the images. It is necessary to present images of higher quality and size. This can be considered an additional weak point.

The paper with opportune modifications can be presented to reconsideration.

Author Response

This paper reports the characterization results of pre-refinement and heat treatment of Al-Si-Mg alloy. This topic is interesting and worthy to be investigated. However, this paper reports just simple characterization results without any scientific discussion based on thermodynamics or diffusion. This paper reports the phase equilibria, which must be discussed with phase diagram study.

However, no suitable phase diagram was suggested. In my opinion, a characterization of the Al-Si-Mg alloy after refinement and heat treatment should be performed to better highlight the grain evolution or in alternative EBSD maps. In this case, it is possible to have information’s concerning the grain shape and size. This will contribute to obtaining strong conclusions and highlights of work.

Response: Thanks for the Reviewer's kind suggestion. We have added the calculation of phase diagram and the description of macroscopic grain structure in the revised draft.

The equilibrium solidification phase diagram of Al-10Si-0.48Mg has been calculated (Fig.1). The precipitation temperatures of α-Al, Si and Mg2Si phases are 593.63, 574.57 and 506.94oC, respectively. The liquid phase vanishing temperature is 565.13 oC. The percentage of each equilibrium phase at room temperature is as follows: 89.829% α-Al, 9.373% Si, and 0.802% Mg2Si. Considering that the melting point of pure aluminum is 667℃, the melting temperature of pure aluminum is set at 675℃. The temperatures of the solid solution treatment and aging treatment are 500 °C and 175 °C, respectively.

Fig. 1 The equilibrium solidification phase diagram of Al-10Si-0.48Mg.

The macrostructures of Al-10Si-0.48Mg alloy after different heat treatment as shown in Fig. 2. The grain size distribution is uniform, and the grain size of the pre-refinement sample is significantly smaller than that of the other two treatments. The grains of the pre-refinement samples show fine equiaxed grains, the grains of the post-refinement samples show coarse equiaxed grains, and the grains of the non-refinement samples show coarse columnar grains.

Fig. 2 Macrostructures of Al-10Si-0.48Mg alloy after different heat treatment, (a) pre-refinement, (b) post-refinement, (c) non-refinement.

In addition, there are many scratches in the figure 5 (c, d), which demonstrates poor preparation of metallographic samples. It is very difficult to verify the validity of the analysis related to Fig. 6 because of the low quality and size of the images. It is necessary to present images of higher quality and size. This can be considered an additional weak point.

Response: Thanks for the Reviewer's kind suggestion. We re-prepared the metallographic sample shown in Figure 5(c, d) and took SEM photos.

Figure 5 (a) SEM morphology of "pre-refinement" as cast sample, (b) XRD diffraction patterns of "pre-refinement" sample (8+20), (c) SEM morphology of "post-refinement" as cast sample, (d) SEM morphology of "non-refinement" as cast samples.

Reviewer 2 Report

In fact, the authors proposed an improved technological process for grain refinement of Al-Si-Mg alloys. The results presented in the manuscript may have some practical value, but their scientific significance raises questions. There are a significant number of evident problems with the paper, which make it unacceptable for publication in the current form. The comments are listed below.

1. It is incorrect to consider Al-Si alloys with a silicon content of about 10 wt.% as high-silicon alloys. In fact, we are talking about hypoeutectic Al-Si alloys, where the silicon content is not high. This applies to both the title and the body of the manuscript.

2. In the Introduction section, the background of the study is clearly insufficiently detailed. This section contains only 11 references. Weak literature review leads to suspect that the authors may not be sufficiently familiar with modern representations about the subject area. This, in a predictable way, affected the substantiation of the scientific novelty of the study, which is not yet obvious in the presented form.

3. There is nothing new in the fact that quenching modes for Al-Si-Mg alloys for two hours are preferable to eight hours. It is well known that the rate of dissolution of the Mg2Si phase at heating temperatures for quenching is very high, and prolonged exposure is accompanied by undesirable coagulation of silicon particles. The scientific rationality of this part of the paper is highly questionable.

4. Some important methodological data required for reproducibility are not indicated. For example, it was necessary to specify the volumes of melting, the dimensions of the resulting ingots, the positions of sampling or cutting out samples for metallographic analysis, etc. The selected temperature conditions for melting and grain refinement processing (675 C) are not generally accepted for alloys of this group; this decision is not justified or explained. Also, no method is described for determining the average grain size of alloys, etc.

5. In addition to the nominal composition (as shown in Table 1), it would be methodically correct to show the actual chemical compositions of the experimental alloys, including the content of impurity elements. It is not specified whether the Al-10Si-0.48Mg alloy is regulated by any standards. If so, then it would be necessary to indicate its brand and the ranges of the content of alloying and impurity elements according to the standard. By itself, the choice of this alloy for research is not clear and not justified. How general will the results be in this case? Will they be applicable to other hypoeutectic alloys? What are the limits and restrictions of the conclusions drawn?

6. The insufficiency of the presented experimental data does not give a complete scientific picture. For example, Fig. 5 contains selective metallographic images and only one XRD pattern. There is no system approach. SEM, EDS, OM, and XRD should be reported for all variants studied and analyzed in an appropriate way.

7. A significant drawback of the manuscript is the weak discussion of the results obtained. In fact, analysis and discussion are performed only fragmentarily. It is necessary to compare the data obtained by the authors with the previously published studies. The absence of a detailed discussion greatly devalues the results presented. 

Based on the major shortcomings of the paper, this reviewer cannot recommend it for publication.

Author Response

In fact, the authors proposed an improved technological process for grain refinement of Al-Si-Mg alloys. The results presented in the manuscript may have some practical value, but their scientific significance raises questions. There are a significant number of evident problems with the paper, which make it unacceptable for publication in the current form. The comments are listed below.

  1. It is incorrect to consider Al-Si alloys with a silicon content of about 10 wt.% as high-silicon alloys. In fact, we are talking about hypoeutectic Al-Si alloys, where the silicon content is not high. This applies to both the title and the body of the manuscript.

Response: Thanks for the Reviewer's kind suggestion. We have modified "high-silicon alloys" in the manuscript to " hypoeutectic Al-Si alloys".

  1. In the Introduction section, the background of the study is clearly insufficiently detailed. This section contains only 11 references. Weak literature review leads to suspect that the authors may not be sufficiently familiar with modern representations about the subject area. This, in a predictable way, affected the substantiation of the scientific novelty of the study, which is not yet obvious in the presented form.

Response: Thanks for the Reviewer's kind suggestion. In order to enhance the scientific nature of the manuscript, the development of grain refiners and toxicological effects are added.

  1. There is nothing new in the fact that quenching modes for Al-Si-Mg alloys for two hours are preferable to eight hours. It is well known that the rate of dissolution of the Mg2Si phase at heating temperatures for quenching is very high, and prolonged exposure is accompanied by undesirable coagulation of silicon particles. The scientific rationality of this part of the paper is highly questionable.

Response: Thanks for the Reviewer's kind suggestion. The reason why the solution holding time is selected as 2 hours and 8 hours is that the solution holding time in T6 heat treatment process specification is 2-8 hours, so we choose the two end values for a comparative study. However, the goal of our manuscript is still that different ways of adding refiners are the key to determine the material properties.

  1. Some important methodological data required for reproducibility are not indicated. For example, it was necessary to specify the volumes of melting, the dimensions of the resulting ingots, the positions of sampling or cutting out samples for metallographic analysis, etc. The selected temperature conditions for melting and grain refinement processing (675 C) are not generally accepted for alloys of this group; this decision is not justified or explained. Also, no method is described for determining the average grain size of alloys, etc.

Response: Thanks for the Reviewer's kind suggestion. In the "experimental Method" section of the revised draft, we have added some experimental details.

The final smelted aluminum ingot size is Φ40×50mm.

The sampling position of the macro and micro structure figures in the manuscript is 25mm axial. The average grain size value is the average value of a total of 6 grain size values taken at the axial direction of 10, 25, and 40mm and the radial direction of 5, 10, and 15mm.

The equilibrium solidification phase diagram of Al-10Si-0.48Mg has been calculated (Fig.1). The precipitation temperatures of α-Al, Si and Mg2Si phases are 593.63, 574.57 and 506.94oC, respectively. The liquid phase vanishing temperature is 565.13 oC. The percentage of each equilibrium phase at room temperature is as follows: 89.829% α-Al, 9.373% Si, and 0.802% Mg2Si. Considering that the melting point of pure aluminum is 667℃, the melting temperature of pure aluminum is set at 675℃. The temperatures of the solid solution treatment and aging treatment are 500 °C and 175 °C, respectively.

Fig. 1 The equilibrium solidification phase diagram of Al-10Si-0.48Mg.

  1. In addition to the nominal composition (as shown in Table 1), it would be methodically correct to show the actual chemical compositions of the experimental alloys, including the content of impurity elements. It is not specified whether the Al-10Si-0.48Mg alloy is regulated by any standards. If so, then it would be necessary to indicate its brand and the ranges of the content of alloying and impurity elements according to the standard. By itself, the choice of this alloy for research is not clear and not justified. How general will the results be in this case? Will they be applicable to other hypoeutectic alloys? What are the limits and restrictions of the conclusions drawn?

Response: Thanks for the Reviewer's kind suggestion. The alloy composition is based on ZL 104(Chinese grade of aluminum alloy), and the goal is to develop a new material for automotive die casting integration.

  1. The insufficiency of the presented experimental data does not give a complete scientific picture. For example, Fig. 5 contains selective metallographic images and only one XRD pattern. There is no system approach. SEM, EDS, OM, and XRD should be reported for all variants studied and analyzed in an appropriate way.

Response: Thanks for the Reviewer's kind suggestion. According to the results of phase diagram calculation, the phase composition of the system is very simple, only α-Al, Si, and Mg2Si phase. We changed Figure 5(c,d) into SEM photos to be more scientific.

Figure 5 (a) SEM morphology of "pre-refinement" as cast sample, (b) XRD diffraction patterns of "pre-refinement" sample (8+20), (c) SEM morphology of "post-refinement" as cast sample, (d) SEM morphology of "non-refinement" as cast samples.

  1. A significant drawback of the manuscript is the weak discussion of the results obtained. In fact, analysis and discussion are performed only fragmentarily. It is necessary to compare the data obtained by the authors with the previously published studies. The absence of a detailed discussion greatly devalues the results presented.

Response: Thanks for the Reviewer's kind suggestion. We add a comparison of the strength and toughness properties with ZL 104 aluminum alloy at the end of "part 4.2" in the manuscript to show the advantages of the new alloy.

The main mechanical properties of ZL104 aluminum alloy are as follows: tensile strength greater than 195MPa, elongation greater than 1.5%, hardness greater than 65HB. In conclusion, Al-10Si-0.48Mg alloy can achieve better mechanical properties than ZL 104 alloy by "pre-refinement" and "2+10" heat treatment process.

Round 2

Reviewer 1 Report

The authors corrected the manuscript according to some of the reviewer's comments. However, the results of determining the grain size by the EBSD method are not available. In general, the article can be accepted for publication.

Reviewer 2 Report

The authors made some attempts to correct the manuscript, considering previous critical comments, but the corrections made are clearly not enough to positively evaluate the manuscript. More specifically: 

1. The Introduction section now additionally describes some of the toxic effects of refiners. However, there is still a lack of a description of the research background in relation to the goal and objectives. It is necessary to focus on the description of Si poisoning mechanisms and possible approaches to dealing with these complications. The Introduction does not reflect all relevant fundamental research in the subject area, including those directly related to the subject of the paper. For example, Si poisoning effect on the grain refinement has been the subject of investigations by M. Johnsson in 1994-1996, later by B.J. McKay and P. Schumacher in 2004, later by A.T. Dinsdale in 2006, by J.A. Taylor then by M.A. Easton and D.H. StJohn in 2014, and many others researchers. 

2. The rationale for the selected heat treatment regimes should be detailed in the manuscript and supported by references.

3. Reviewer approves analysis of the equilibrium solidification phase diagram added by authors. However, for a more accurate analysis, it is desirable to supplement the description with the calculation of non-equilibrium crystallization according to the Sheil-Gulliver model in the Thermo-Calc software. 

4. It is not entirely clear what the authors mean by the following: "the goal is to develop a new material for automotive die casting integration"? Are the authors developing new material? But this paper is not about that.

5. The previous note on nominal compositions (Table 1) remains the same. It would be methodically correct to show the actual chemical compositions of the experimental alloys, including the content of impurity elements. Also, it is necessary to indicate alloy's brand and the ranges of the content of alloying and impurity elements according to the standard. 

6. The authors supplemented Fig. 6 with SEM images, which is commendable, as well as the addition of macrostructures (Fig. 7). However, Fig. 6(a) and Fig. 6(c,d) have different magnifications, which makes it impossible to compare them. 

7. As before, this reviewer found the discussion to be rather short and lacking in depth. Theoretical interpretation of experimental data still does not correspond to high quality criteria of Materials journal. It is necessary to explain from generally accepted scientific positions the influence of the suggested decisions on the refinement effect, to provide scientific substantiations of a concept of "pre-refinement". 

8. The scheme in Fig. 1 as presented is difficult to understand. It is desirable to divide it into a larger number of stages, as described in the previous description. In particular, separate stage 1 with the addition of a refiner to the Al-Mg alloy and stage 2 with the addition of silicon. Also show the direction of growth of alpha-aluminum in stage 3. 

9. The manuscript strongly needs to be carefully edited for grammar and syntax.

Overall recommendation: Reconsider after major revision (control missing in some experiments). Reconsideration will be possible only if all the comments indicated are clearly taken into account, primarily targeted at increasing the level of depth in the discussion. 

Author Response

  1. The Introduction section now additionally describes some of the toxic effects of refiners. However, there is still a lack of a description of the research background in relation to the goal and objectives. It is necessary to focus on the description of Si poisoning mechanisms and possible approaches to dealing with these complications. The Introduction does not reflect all relevant fundamental research in the subject area, including those directly related to the subject of the paper. For example, Si poisoning effect on the grain refinement has been the subject of investigations by M. Johnsson in 1994-1996, later by B.J. McKay and P. Schumacher in 2004, later by A.T. Dinsdale in 2006, by J.A. Taylor then by M.A. Easton and D.H. StJohn in 2014, and many others researchers.

Response: Thanks for the Reviewer's kind suggestion. The revised draft added the introduction of the research background of silicon poisoning mechanism.

"Schumacher and McKay found that the formation of TiSi2 on basal faces of TiB2, reducing the nucleation area and number of active sites for Al [11]. The reduction in a-Al growth restriction and the formation of silicide phases before solidification of a-Al due to a strong exothermic interaction between titanium and silicon, which is the main mechanism of Si poisoning [12]. In fact, the small degree of grain refinement by additions of eutectic-forming elements (Cu, Mg and Si) is mainly attributed to their segregating power, however, they can not form nucleant particles [13 ].The nucleation work of α-Al on its surface increases, and the grain refinement effect decreases [14,15]. It should be noted that poisoning occurs whether or not it is grain refined with AlTiB master alloys [16]. Therefore, we must find a way to circumvent the influence of Si on the nucleation and growth of α-Al."

  1. The rationale for the selected heat treatment regimes should be detailed in the manuscript and supported by references.

Response: Thanks for the Reviewer's kind suggestion. The revised draft added the introduction of the selected heat treatment regimes.

"Solution heat treatment at 500-530 °C for 2-8h after water quenching and artificial aging at 170-210 °C for 2-20h were the commonly used heat treatment to improve the mechanical properties of the alloys [24,25]. Likewise, the T6 heat treatment process of solid solution treatment followed by aging treatment was carried out for the three kinds of as-cast alloys."

  1. Reviewer approves analysis of the equilibrium solidification phase diagram added by authors. However, for a more accurate analysis, it is desirable to supplement the description with the calculation of non-equilibrium crystallization according to the Sheil-Gulliver model in the Thermo-Calc software.

Response: Thanks for the Reviewer's kind suggestion. The solidification paths of Al-10Si-0.48Mg by equilibrium and Scheil schemes (non-equilibrium solidification) were calculated with the trial version of Thermo_Calc.

Figure 2 (a) The equilibrium solidification phase diagram of Al-10Si-0.48Mg, (b) Calculated solidification paths of Al-10Si-0.48Mg by equilibrium and Scheil schemes.

"We use the results of thermodynamic calculation to help use design the key parmeters during the pre-refinement and the following heat treatment process. The equilibrium solidification phase diagram of Al-10Si-0.48Mg with the built-in database was calculated with the trial version of Thermo_Calc (Figure 2(a), the database was restored according to reference [23]). The precipitation temperatures of α-Al, Si and Mg2Si phases are 593.63, 574.57 and 506.94oC, respectively. The liquid phase vanishing temperature is 565.13oC. The percentage of each equilibrium phase at room temperature is as follows: 89.829% α-Al, 9.373% Si, and 0.802% Mg2Si. The solidification paths of Al-10Si-0.48Mg by equilibrium and Scheil schemes (non-equilibrium solidification) were calculated with the trial version of Thermo_Calc (Figure 2(b)). The Mg2Si precipitates at 559℃ of non-equilibrium solidification. The final solidification temperature is 6.5oC lower than that of equilibrium solidification."

  1. It is not entirely clear what the authors mean by the following: "the goal is to develop a new material for automotive die casting integration"? Are the authors developing new material? But this paper is not about that.

Response: This is our expression is not clear enough caused by the misunderstanding. The revised draft contains new instructions.

"The original purpose of choosing Al-10Si-0.48Mg alloy was to develop a new material for automotive die casting integration. But the Si content is more than 3.5 wt.%, the conventional method of adding Al-Ti-B refiner can not produce the refining effect. In view of this, we hope to change the melting order of refiners and alloying elements to achieve the traditional refiners still play the role of refining."

  1. The previous note on nominal compositions (Table 1) remains the same. It would be methodically correct to show the actual chemical compositions of the experimental alloys, including the content of impurity elements. Also, it is necessary to indicate alloy's brand and the ranges of the content of alloying and impurity elements according to the standard.

Response: Thanks for the Reviewer's kind suggestion. Table 1 is really unnecessary and deleted.

  1. The authors supplemented Fig. 6 with SEM images, which is commendable, as well as the addition of macrostructures (Fig. 7). However, Fig. 6(a) and Fig. 6(c,d) have different magnifications, which makes it impossible to compare them.

Response: Thanks for the Reviewer's kind suggestion. It is ideal to be able to compare phase and microstructure morphology with the same magnifications. The purpose of Fig. 6 is to illustrate the phase and microstructure morphology of the alloys prepared in different ways. When we take SEM photos, we mainly consider to clearly show the alloy phase and microstructure appearance of different grain sizes, but do not pursue the unification of magnification.

  1. As before, this reviewer found the discussion to be rather short and lacking in depth. Theoretical interpretation of experimental data still does not correspond to high quality criteria of Materials journal. It is necessary to explain from generally accepted scientific positions the influence of the suggested decisions on the refinement effect, to provide scientific substantiations of a concept of "pre-refinement".

Response: Thanks for the Reviewer's kind suggestion. This manuscript mainly introduces the effectiveness of the pre-refinement method through a comparative study. The principle of this method is as shown in Figure 1 of the manuscript. The heteronucleation of Al3Ti compounds is preferentially formed, and Si element is finally added to prevent the generation of Ti-Si compounds, so as to achieve the purpose of grain refinement. As for its deep microscopic mechanism needs further study.

  1. The scheme in Fig. 1 as presented is difficult to understand. It is desirable to divide it into a larger number of stages, as described in the previous description. In particular, separate stage 1 with the addition of a refiner to the Al-Mg alloy and stage 2 with the addition of silicon. Also show the direction of growth of alpha-aluminum in stage 3.

Response: Thanks for the Reviewer's kind suggestion. We have modified Figure 1 to add intermediate steps to make it easier to understand how it works.

Figure 1 Schematic diagram of "pre-refinement" design concept.

  1. The manuscript strongly needs to be carefully edited for grammar and syntax.

Response: Thanks for the Reviewer's kind suggestion. The manuscript has re-edited for grammar and syntax.
